# Improved Training Speed, Accuracy, and Data Utilization via Loss Function Optimization

## Abstract

As the complexity of neural network models has grown, it has become increasingly important to optimize their design automatically through metalearning. Methods for discovering hyperparameters, topologies, and learning rate schedules have lead to significant increases in performance. This paper shows that loss functions can be optimized with metalearning as well, and result in similar improvements. The method, Genetic Loss-function Optimization (GLO), discovers loss functions *de novo*, and optimizes them for a target task. Leveraging techniques from genetic programming, GLO builds loss functions hierarchically from a set of operators and leaf nodes. These functions are repeatedly recombined and mutated to find an optimal structure, and then a covariance-matrix adaptation evolutionary strategy (CMA-ES) is used to find optimal coefficients. Networks trained with GLO loss functions are found to outperform the standard cross-entropy loss on standard image classification tasks. Training with these new loss functions requires fewer steps, results in lower test error, and allows for smaller datasets to be used. Loss function optimization thus provides a new dimension of metalearning, and constitutes an important step towards AutoML.

## 1 Introduction

Much of the power of modern neural networks originates from their complexity, i.e., number of parameters, hyperparameters, and topology. This complexity is often beyond human ability to optimize, and automated methods are needed. An entire field of metalearning has emerged recently to address this issue, based on various methods such as gradient descent, simulated annealing, reinforcement learning, Bayesian optimization, and evolutionary computation (EC) (Elsken et al., 2018).

While a wide repertoire of work now exists for optimizing many aspects of neural networks, the dynamics of training are still usually set manually without concrete, scientific methods. Training schedules, loss functions, and learning rates all affect the training and final functionality of a neural network. Perhaps they could also be optimized through metalearning?

The goal of this paper is to verify this hypothesis, focusing on optimization of loss functions. A general framework for loss function metalearning, covering both novel loss function discovery and optimization, is developed and evaluated experimentally. This framework, Genetic Loss-function Optimization (GLO), leverages Genetic Programming to build loss functions represented as trees, and subsequently a Covariance-Matrix Adaptation Evolution Strategy (CMA-ES) to optimize their coefficients.

EC methods were chosen because EC is arguably the most versatile of the metalearning approaches. EC, being a type of population-based search method, allows for extensive exploration, which often results in creative, novel solutions (Lehman et al., 2018). EC has been successful in hyperparameter optimization and architecture design in particular (Miikkulainen et al., 2019; Stanley et al., 2019; Real et al., 2019; Loshchilov & Hutter, 2016). It has also been used to discover mathematical formulas to explain experimental data (Schmidt & Lipson, 2009). It is, therefore, likely to find creative solutions in the loss-function optimization domain as well.

Indeed, on the MNIST image classification benchmark, GLO discovered a surprising new loss function, named Baikal for its shape. This function performs very well, presumably by establishing an implicit regularization effect. Baikal outperforms the standard cross-entropy loss in terms of training

speed, final accuracy, and data requirements. Furthermore, Baikal was found to transfer to a more complicated classification task, CIFAR-10, while carrying over its benefits.

At first glance, Baikal behaves rather unintuitively; loss does not decrease monotonically as a network's predictions become more correct. Upon further analysis, Baikal was found to perform implicit regularization, which caused this effect. Specifically, by preventing the network from being too confident in its predictions, training was able to produce a more robust model. This finding was surprising and encouraging, since it means that GLO is able to discover loss functions that train networks that are more generalizable and overfit less.

The next section reviews related work in metalearning and EC, to help motivate the need for GLO. Following this review, GLO is described in detail, along with the domains upon which it has been evaluated. The subsequent sections present the experimental results, including an analysis of the loss functions that GLO discovers.

## 2 RELATED WORK

In addition to hyperparameter optimization and neural architecture search, new opportunities for metalearning have recently emerged. In particular, learning rate scheduling and adaptation can have a significant impact on a model's performance. Learning rate schedules determine how the learning rate changes as training progresses. This functionality tends to be encapsulated away in practice by different gradient-descent optimizers, such as AdaGrad (Duchi et al., 2011) and Adam (Kingma & Ba, 2014). While the general consensus has been that monotonically decreasing learning rates yield good results, new ideas, such as cyclical learning rates (Smith, 2017), have shown promise in learning better models in fewer epochs.

Metalearning methods have also been recently developed for data augmentation, such as AutoAugment (Cubuk et al., 2018), a reinforcement learning based approach to find new data augmentation policies. In reinforcement learning tasks, EC has proven a successful approach. For instance, in evolving policy gradients (Houthooft et al., 2018), the policy loss is not represented symbolically, but rather as a neural network that convolves over a temporal sequence of context vectors. In reward function search (Niekum et al., 2010), the task is framed as a genetic programming problem, leveraging PushGP (Spector et al., 2001).

In terms of loss functions, a generalization of the $L^2$ loss was proposed with an adaptive loss parameter (Barron, 2017). This loss function is shown to be effective in domains with multivariate output spaces, where robustness might vary across between dimensions. Specifically, the authors found improvements in Variational Autoencoder (VAE) models, unsupervised monocular depth estimation, geometric registration, and clustering.

Additionally, work has found promise in moving beyond the standard cross-entropy loss for classification (Janocha & Czarnecki, 2017). $L^1$ and $L^2$ losses were found to have useful probabilistic properties. The authors found certain loss functions to be more resilient to noise than the cross-entropy loss.

Notably, no existing work in the metalearning literature automatically optimizes loss functions for neural networks. As shown in this paper, evolutionary computation can be used in this role to improve neural network performance, gain a better understanding of the processes behind learning, and help reach the ultimate goal of fully automated learning.

## 3 THE GLO APPROACH

The task of finding and optimizing loss functions can be framed as a functional regression problem. GLO accomplishes this through the following high-level steps (shown in Figure 1): **(1) loss function discovery:** using approaches from genetic programming, a genetic algorithm builds new candidate loss functions, and **(2) coefficient optimization:** to further optimize a specific loss function, a covariance-matrix adaptation evolutionary strategy (CMA-ES) is leveraged to optimize coefficients.

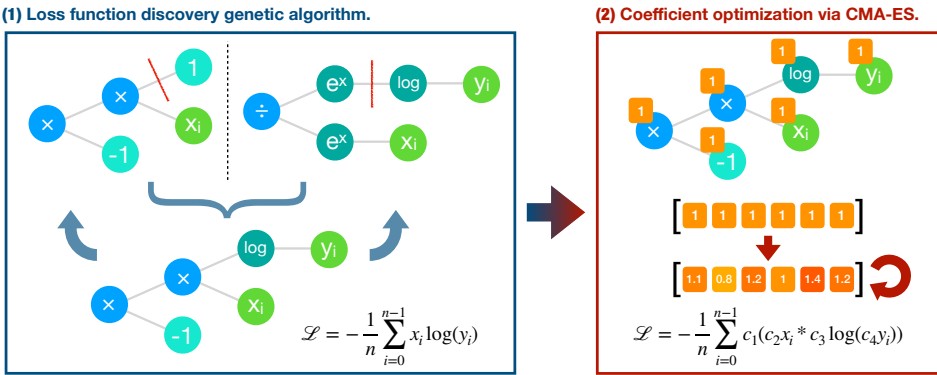

Figure 1: Genetic Loss-function Optimization (GLO) overview. A genetic algorithm constructs candidate loss functions as trees. The best loss functions from this set then has its coefficients optimized using CMA-ES. GLO loss functions are able to train models more quickly and more accurately.

### 3.1 LOSS FUNCTION DISCOVERY

GLO uses a population-based search approach, inspired by genetic programming, to discover new optimized loss function candidates. Under this framework, loss functions are represented as trees within a genetic algorithm. Trees are a logical choice to represent functions due to their hierarchical nature. The loss function search space is defined by the following tree nodes:

**Unary Operators:** $\log(\circ), \circ^2, \sqrt{\circ}$
**Binary Operators:** $+, *, -, \div$
**Leaf Nodes:** $x, y, 1, -1$, where $x$ represents a true label, and $y$ represents a predicted label.

The search space is further refined by automatically assigning a fitness of 0 to trees that do not contain both at least one $x$ and one $y$. Generally, a loss function's fitness within the genetic algorithm is the validation performance of a network trained with that loss function. To expedite the discovery process, and encourage the invention of loss functions that make learning faster, training does not proceed to convergence. Unstable training sessions that result in NaN values are assigned a fitness of 0. Fitness values are cached to avoid needing to retrain the same network twice. These cached values are each associated with a canonicalized version of their corresponding tree, resulting in fewer required evaluations.

The initial population is composed of randomly generated trees with a maximum depth of 2. Recursively starting from the root, nodes are randomly chosen from the allowable operator and leaf nodes using a weighting (where $\log(\circ), x, y$ are three times as likely and $\sqrt{\circ}$ is two times as likely as $+, *, -, \div, 1, -1$). This weighting can impart a bias and prevent, for example, the integer 1 from occurring too frequently. The genetic algorithm has a population size of 80, incorporates elitism with six elites per generation, and uses roulette sampling.

Recombination is accomplished by randomly splicing two trees together. For a given pair of parent trees, a random element is chosen in each as a crossover point. The two subtrees, whose roots are the two crossover points, are then swapped with each other. Figure 1 presents an example of this method of recombination. Both resultant trees become part of the next generation. Recombination occurs with a probability of $80\%$.

To introduce variation into the population, the genetic algorithm has the following mutations, applied in a bottom-up fashion:

- Integer scalar nodes are incremented or decremented with a $5\%$ probability.
- Nodes are replaced with a weighted-random node with the same number of children with a $5\%$ probability.
- Nodes (and their children) are deleted and replaced with a weighted-random leaf node with a $5\% * 50\% = 2.5\%$ probability.

- Leaf nodes are deleted and replaced with a weighted-random element (and weighted-random leaf children if necessary) with a $5\% * 50\% = 2.5\%$ probability.

Combined, the iterative sampling, recombination, and mutation of trees within the population leads to the discovery of new loss functions which maximize fitness.

## 3.2 COEFFICIENT OPTIMIZATION

Loss functions found by the above genetic algorithm can all be thought of having unit coefficients for each node in the tree. This set of coefficients can be represented as a vector with dimensionality equal to the number of nodes in a loss function's tree. The number of coefficients can be reduced by pruning away coefficients that can be absorbed by others (e.g., $3 (5x + 2y) = 15x + 6y$). The coefficient vector is optimized independently and iteratively using a covariance-matrix adaptation evolutionary strategy (CMA-ES) (Hansen & Ostermeier, 1996). The specific variant of CMA-ES that GLO uses is $(\mu/\mu, \lambda)$-CMA-ES (Hansen & Ostermeier, 2001), which incorporates weighted rank-$\mu$ updates (Hansen & Kern, 2004) to reduce the number of objective function evaluations that are needed. The implementation of GLO presented in this paper uses an initial step size $\sigma = 1.5$. As in the discovery phase, the objective function is the network's performance on a validation dataset after a shortened training period.

## 4 EXPERIMENTAL EVALUATION

This section provides an experimental evaluation of GLO, on the MNIST and CIFAR-10 image classification tasks. Baikal, a GLO loss function found on MNIST, is presented and evaluated in terms of its resulting testing accuracy, training speed, training data requirements, and transferability to CIFAR-10. Implementation details are presented in the appendix in Section A.1.

### 4.1 TARGET TASKS

Experiments on GLO are performed using two popular image classification datasets, MNIST Handwritten Digits (LeCun et al., 1998) and CIFAR-10 (Krizhevsky & Hinton, 2009). Both datasets, with MNIST in particular, are well understood, and relatively quick to train. The choice of these datasets allowed rapid iteration in the development of GLO and allowed time for more thorough experimentation. The selected model architectures are simple, since achieving state-of-the-art accuracy on MNIST and CIFAR-10 is not the focus of this paper, rather the improvements brought about by using a GLO loss function are. More information on the datasets, along with their corresponding architectures and experimental setup is provided in the appendix, under Section A.2.

Both of these tasks, being classification problems, are traditionally framed with the standard cross-entropy loss (sometimes referred to as the log loss): $\mathcal{L}_{\text{Log}} = -\frac{1}{n} \sum_{i=0}^{n-1} x_i \log(y_i)$, where $\mathbf{x}$ is sampled from the true distribution, $\mathbf{y}$ is from the predicted distribution, and $n$ is the number of classes. The cross-entropy loss is used as a baseline in this paper's experiments.

### 4.2 THE BAIKAL LOSS FUNCTION

The most notable loss function that GLO discovered against the MNIST dataset (with 2,000-step training for candidate evaluation) is the Baikal loss, named due to its similarity to the bathymetry of Lake Baikal when its binary variant is plotted in 3D (Section 5.1):

$$\mathcal{L}_{\text{Baikal}} = -\frac{1}{n} \sum_{i=0}^{n-1} \log(y_i) - \frac{x_i}{y_i} \, , \tag{1}$$

where $x$ is from the true distribution, $y$ is from the predicted distribution, and $n$ is the number of classes. Additionally, after coefficient optimization, GLO arrived at the following version of the Baikal loss:

$$\mathcal{L}_{\text{BaikalCMA}} = -\frac{1}{n} \sum_{i=0}^{n-1} 2.7279 \left( 0.9863 * \log(1.5352 * y_i) - 1.8158 \, \frac{x_i}{y_i} \right) \, . \tag{2}$$

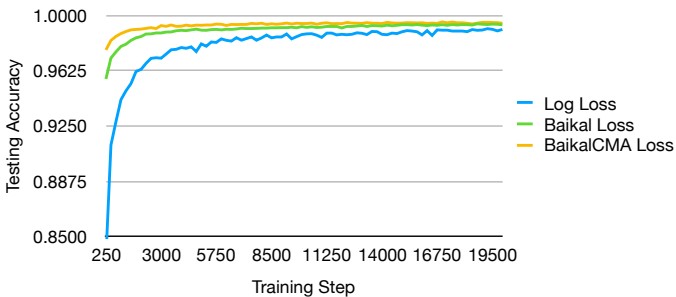

Figure 3: Training curves for different loss functions on MNIST. Baikal and BaikalCMA result in faster and smoother training compared to the cross-entropy loss.

This loss function, BaikalCMA, was selected for having the highest validation accuracy out of the population. The Baikal and BaikalCMA loss functions had validation accuracies at 2,000 steps equal to 0.9838 and 0.9902, respectively. For comparison, the cross-entropy loss had a validation accuracy at 2,000 steps of 0.9700. Models trained with the Baikal loss on MNIST and CIFAR-10 (to test transfer) are the primary vehicle for validating GLO's efficacy, as detailed in subsequent sections.

### 4.3 TESTING ACCURACY

Figure 2 shows the increase in testing accuracy that Baikal and BaikalCMA provide on MNIST over models trained with the cross-entropy loss. Over 10 trained models each, the mean testing accuracies for cross-entropy loss, Baikal, and BaikalCMA were 0.9899, 0.9933, and **0.9947**, respectively.

This increase in accuracy from Baikal over cross-entropy loss is found to be statistically significant, with a $p$-value of $2.4 \times 10^{-11}$, in a heteroscedastic, two-tailed T-test, with 10 samples from each distribution. With the same significance test, the increase in accuracy from BaikalCMA over Baikal was found to be statistically significant, with a $p$-value of $8.5045 \times 10^{-6}$.

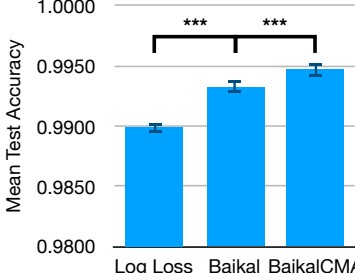

Figure 2: Mean testing accuracy on MNIST, $n = 10$. Both Baikal and BaikalCMA provide statistically significant improvements to testing accuracy over the cross-entropy loss.

### 4.4 TRAINING SPEED

Training curves for networks trained with the cross-entropy loss, Baikal, and BaikalCMA are shown in Figure 3. Each curve represents 80 testing dataset evaluations spread evenly (i.e., every 250 steps) throughout 20,000 steps of training on MNIST. Networks trained with Baikal and BaikalCMA both learn significantly faster than the cross-entropy loss. These phenomena make Baikal a compelling loss function for fixed time-budget training, where the improvement in resultant accuracy over the cross-entropy loss becomes most evident.

### 4.5 TRAINING DATA REQUIREMENTS

Figure 4 provides an overview of the effects of dataset size on networks trained with cross-entropy loss, Baikal, and BaikalCMA. For each training dataset portion size, five individual networks were trained for each loss function.

The degree by which Baikal and BaikalCMA outperform cross-entropy loss increases as the training dataset becomes smaller. This provides evidence of less overfitting when training a network with Baikal or BaikalCMA. As expected, BaikalCMA outperforms Baikal at all tested dataset sizes. The size of this improvement in accuracy does not grow as significantly as the improvement over cross-entropy loss, leading to the belief that the overfitting characteristics of Baikal and BaikalCMA

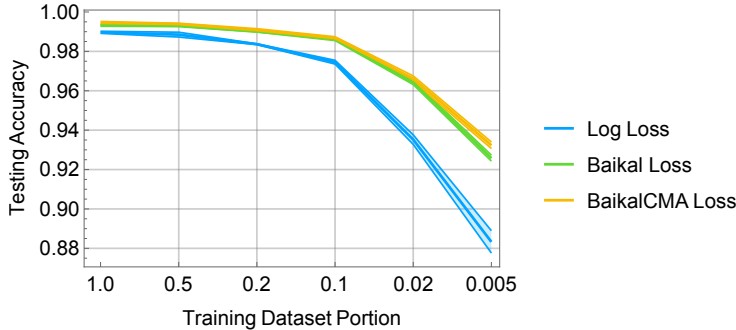

Figure 4: Sensitivity to different dataset sizes for different loss functions on MNIST. For each size, $n = 5$. Baikal and BaikalCMA increasingly outperform the cross-entropy loss on small datasets, providing evidence of reduced overfitting.

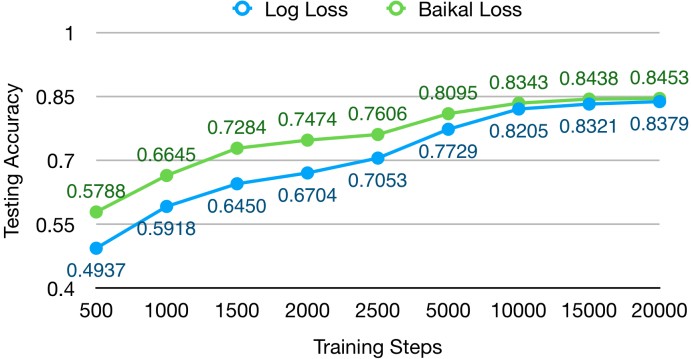

Figure 5: Testing accuracy across varying training steps on CIFAR-10. The Baikal loss, which has been transferred from MNIST, outperforms the cross-entropy loss on all training durations.

are very similar. Ostensibly, one could run the optimization phase of GLO on a reduced dataset specifically to yield a loss function with better performance than BaikalCMA on small datasets.

## 4.6 LOSS FUNCTION TRANSFER TO CIFAR-10

Figure 5 presents a collection of 18 separate tests of the cross-entropy loss and Baikal applied to CIFAR-10. Baikal is found to outperform cross-entropy across all training durations, with the difference becoming more prominent for shorter training periods. These results present an interesting use case for GLO, where a loss function that is found on a simpler dataset can be transferred to a more complex dataset while still maintaining performance improvements. This faster training provides a particularly persuasive argument for using GLO loss functions in fixed time-budget scenarios.

## 5 WHAT MAKES BAIKAL WORK?

This section presents a symbolic analysis of the Baikal loss function, followed by experiments that attempt to elucidate why Baikal works better than the cross-entropy loss. A likely explanation is that Baikal results in implicit regularization, reducing overfitting.

## 5.1 BINARY CLASSIFICATION

Loss functions used on the MNIST dataset, a 10-dimensional classification problem, are difficult to plot and visualize graphically. To simplify, loss functions are analyzed in the context of binary

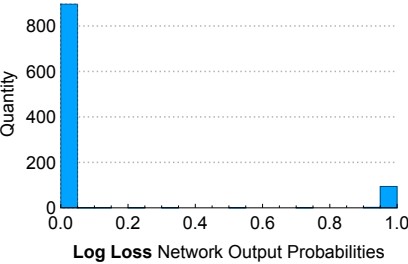 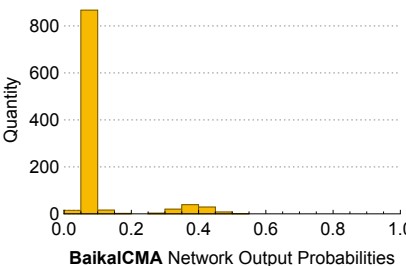

Figure 7: Output probabilities of networks trained with cross-entropy loss and BaikalCMA. With BaikalCMA, the peaks are shifted away from extreme values and more spread out, indicating implicit regularization. The BaikalCMA histogram matches that from a network trained with a confidence regularizer (Pereyra et al., 2017).

classification, with $n = 2$, the Baikal loss expands to

$$\mathcal{L}_{\text{Baikal2D}} = -\frac{1}{2}\left(\log(y_0) - \frac{x_0}{y_0} + \log(y_1) - \frac{x_1}{y_1}\right). \tag{3}$$

Since vectors $\mathbf{x}$ and $\mathbf{y}$ sum to 1, by consequence of being passed through a softmax function, for binary classification $\mathbf{x} = \langle x_0, 1 - x_0 \rangle$ and $\mathbf{y} = \langle y_0, 1 - y_0 \rangle$. This constraint simplifies the binary Baikal loss to the following function of two variables ($x_0$ and $y_0$):

$$\mathcal{L}_{\text{Baikal2D}} \propto -\log(y_0) + \frac{x_0}{y_0} - \log(1 - y_0) + \frac{1 - x_0}{1 - y_0}. \tag{4}$$

This same methodology can be applied to the cross-entropy loss and BaikalCMA.

In practice, true labels are assumed to be correct with certainty, thus, $x_0$ is equal to either 0 or 1. The specific case where $x_0 = 1$ is plotted in Figure 6 for the cross-entropy loss, Baikal, and BaikalCMA. The cross-entropy loss is shown to be monotonically decreasing, while Baikal and BaikalCMA counterintuitively show an increase in the loss value as the predicted label, $y_0$, approaches the true label $x_0$. This unexpected increase allows the loss functions to prevent the model from becoming too confident in its output predictions, thus providing a form of regularization. Section 5.2 provides reasoning for this unexplained result.

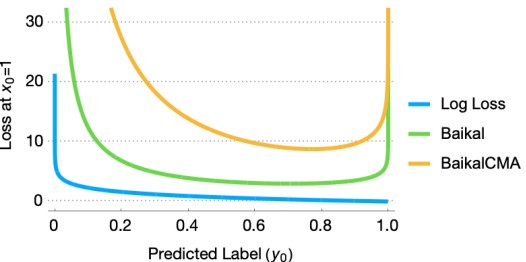

Figure 6: Binary classification loss functions at $x_0 = 1$. Correct predictions lie on the right side of the graph, and incorrect ones on the left. The log loss decreases monotonically, while Baikal and BaikalCMA present counterintuitive, sharp increases in loss as predictions, approach the true label. This phenomenon provides regularization by preventing the model from being too confident in its predictions.

As also seen in Figure 6, the minimum for the Baikal loss where $x_0 = 1$ lies near 0.71, while the minimum for the BaikalCMA loss where $x_0 = 1$ lies near 0.77. This minimum, along with the more pronounced slope around $x_0 = 0.5$ is likely a reason why BaikalCMA performs better than Baikal.

## 5.2 IMPLICIT REGULARIZATION

The Baikal and BaikalCMA loss functions are surprising in that they incur a high loss when the output is very close to the correct value (as illustrated in Figure 6). Although at first glance this behavior is counterintuitive, it may provide an important advantage. The outputs of a trained network will not be exactly correct, although they are close, and therefore the network is less likely to overfit. Thus, these loss functions provide an implicit form of regularization, enabling better generalization.

This effect is similar to that of the confidence regularizer (Pereyra et al., 2017), which penalizes low-entropy prediction distributions. The bimodal distribution of output probabilities that results from confidence regularization is nearly identical to that of a network trained with BaikalCMA. Histograms of these distributions on the test dataset for cross-entropy and BaikalCMA networks, after 15,000 steps of training on MNIST, are shown in Figure 7. The abscissae in Figures 6 and 7 match, making it clear how the distribution for BaikalCMA has shifted away from the extreme values. The improved behavior under small-dataset conditions described in Section 4.5 further supports implicit regularization; less overfitting was observed when using Baikal and BaikalCMA compared to the cross-entropy loss.

Notably, the implicit regularization provided by Baikal and BaikalCMA complements the different types of regularization already present in the trained networks. As detailed in Section A.2, MNIST networks are trained with dropout (Hinton et al., 2012), and CIFAR-10 networks are trained with $L^2$ weight decay and local response normalization (Krizhevsky et al., 2012), yet Baikal is able to improve performance further.

## 6  DISCUSSION AND FUTURE WORK

This paper proposes loss function discovery and optimization as a new form of metalearning, and introduces an evolutionary computation approach to it. GLO was evaluated experimentally in the image classification domain, and discovered a surprising new loss function, Baikal. Experiments showed substantial improvements in accuracy, convergence speed, and data requirements. Further analysis suggested that these improvements result from implicit regularization that reduces overfitting to the data. This regularization complements the existing regularization in trained networks.

In the future, GLO can be applied to other machine learning datasets and tasks. The approach is general, and could result in discovery of customized loss functions for different domains, or even specific datasets. One particularly interesting domain is generative adversarial networks (GANs). Significant manual tuning is necessary in GANs to ensure that the generator and discriminator networks learn harmoniously. GLO could find co-optimal loss functions for the generator and discriminator networks in tandem, thus making GANs more powerful, robust, and easier to implement.

GAN optimization is an example of co-evolution, where multiple interacting solutions are developed simultaneously. GLO could leverage co-evolution more generally: for instance, it could be combined with techniques like CoDeepNEAT (Miikkulainen et al., 2019) to learn jointly-optimal network structures, hyperparameters, learning rate schedules, data augmentation, and loss functions simultaneously. Such an approach requires significant computing power, but may also discover and utilize interactions between the design elements that result in higher complexity and better performance than is currently possible.

## 7  CONCLUSION

This paper proposes Genetic Loss-function Optimization (GLO) as a general framework for discovering and optimizing loss functions for a given task. A surprising new loss function, Baikal, was discovered in the experiments, and shown to outperform the cross-entropy loss on MNIST and CIFAR-10 in terms of accuracy, training speed, and data requirements. Further analysis suggested that Baikal's improvements result from implicit regularization that reduces overfitting to the data. GLO can be combined with other aspects of metalearning in the future, paving the way to robust and powerful AutoML.

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

## A  APPENDIX

### A.1  IMPLEMENTATION DETAILS

Due to the large number of partial training sessions that are needed for both the discovery and optimization phases, training is distributed across the network to a cluster of dedicated machines that use Condor (Thain et al., 2005) for scheduling. Each machine in this cluster has one NVIDIA GeForce GTX Titan Black GPU and two Intel Xeon E5-2603 (4 core) CPUs running at 1.80GHz with 8GB of memory. Training itself is implemented with TensorFlow (Abadi et al., 2016) in Python. The primary components of GLO (i.e., the genetic algorithm and CMA-ES) are implemented in Swift. These components run centrally on one machine and asynchronously dispatch work to the Condor cluster over SSH.

### A.2  EXPERIMENTAL SETUP

The following two sections detail the experimental setup that was used for the evaluation presented in this paper.

### A.2.1  MNIST

The first target task used for evaluation was the MNIST Handwritten Digits dataset (LeCun et al., 1998), a widely used dataset where the goal is to classify $28 \times 28$ pixel images as one of ten digits. The MNIST dataset has 55,000 training samples, 5,000 validation samples, and 10,000 testing samples.

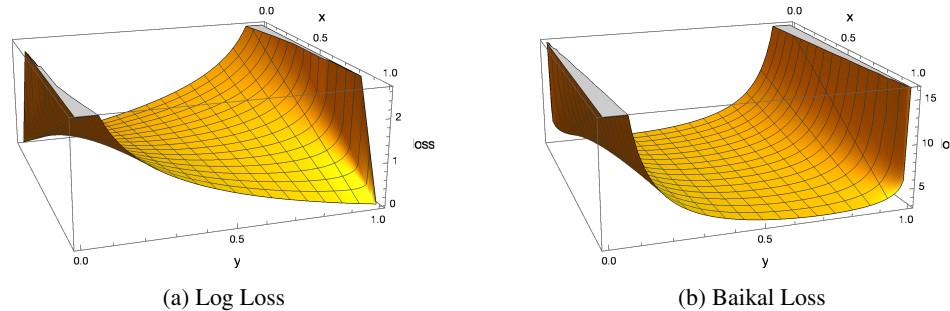

(a) Log Loss  (b) Baikal Loss

Figure 8: Loss function surface plots for binary classification. $y$ is the prediction for one class, and $x$ is the true label for that class. Baikal is channel-shaped, while the log loss has a saddle-like shape.

A simple CNN architecture with the following layers is used: (1) $5 \times 5$ convolution with 32 filters, (2) $2 \times 2$ stride-2 max-pooling, (3) $5 \times 5$ convolution with 64 filters, (4) $2 \times 2$ stride-2 max-pooling, (5) 1024-unit fully-connected layer, (6) a dropout layer (Hinton et al., 2012) with 40% dropout probability, and (7) a softmax layer. ReLU (Nair & Hinton, 2010) activations are used. Training uses stochastic gradient descent (SGD) with a batch size of 100, a learning rate of 0.01, and, unless otherwise specified, occurred over 20,000 steps.

Several experiments tested various learning rate values across a handful of orders-of-magnitude to arrive at the step size used in the paper. For the baseline, this step size provided the highest accuracy.

### A.2.2 CIFAR-10

To further validate GLO, the more challenging CIFAR-10 dataset (Krizhevsky & Hinton, 2009), a popular dataset of small, color photographs in ten classes, was used as a medium to test the transferability of loss functions found on a different domain. CIFAR-10 consists of 50,000 training samples, and 10,000 testing samples.

A simple CNN architecture, taken from (Gonzalez et al., 2019) (and itself inspired by AlexNet (Krizhevsky et al., 2012)), with the following layers is used: (1) $5 \times 5$ convolution with 64 filters and ReLU activations, (2) $3 \times 3$ max-pooling with a stride of 2, (3) local response normalization (Krizhevsky et al., 2012) with $k = 1, \alpha = 0.001/9, \beta = 0.75$, (4) $5 \times 5$ convolution with 64 filters and ReLU activations, (5) local response normalization with $k = 1, \alpha = 0.001/9, \beta = 0.75$, (6) $3 \times 3$ max-pooling with a stride of 2, (7) 384-unit fully-connected layer with ReLU activations, (8) 192-unit fully-connected, linear layer, and (9) a softmax layer.

Inputs to the network are sized $24 \times 24 \times 3$, rather than $32 \times 32 \times 3$ as provided in the dataset; these smaller sized inputs enable more sophisticated data augmentation. To force the network to learn better spatial invariance, random $24 \times 24$ croppings are selected from each full-size image, randomly flipped longitudinally, randomly lightened or darkened, and their contrast is randomly perturbed. Furthermore, to attain quicker convergence, an image's mean pixel value and variance are subtracted and divided, respectively, from the whole image during training and evaluation. CIFAR-10 networks were trained with SGD, $L^2$ regularization with a weight decay of 0.004, a batch size of 1024, and an initial learning rate of 0.05 that decays by a factor of 0.1 every 350 epochs.

Several experiments tested various initial learning rate values across a handful of orders-of-magnitude to arrive at the step size used in the paper. For the baseline, this initial learning rate provided the highest accuracy.

### A.3 BINARY CLASSIFICATIONS SURFACE PLOTS

When plotted in three-dimensions, as in Figure 8, the binary cross-entropy and Baikal loss functions can be observed to have characteristic surfaces. The shape of Baikal's surface, and its similarity to the bathymetry of Lake Baikal, is where it gets its name. Note that the case plotted in Figure 6 is equivalent to the front "slice" of the surface plots in Figure 8.

