# OpenReview forum: "Improved Training Speed, Accuracy, and Data Utilization via Loss Function Optimization"
_ICLR.cc/2020/Conference — Reject_

### Official Review · AnonReviewer3 · 2019-10-16
**Official Blind Review #3**

**Rating:** 3

**Review:**

The authors present a framework to perform meta-learning on the loss used for
training. They introduce the Baikal loss, obtained using the MNIST dataset, and
BaikalCMA where the coefficients have been tuned. The evaluation of these loss
functions is performed on the MNIST and CIFAR-10, and according to the results
they converge faster, towards lower test error and need fewer samples to obtain
results similar to the cross-entropy loss.

The claims are clearly stated and the framework is detailed, the experiments
cover all the potential benefits of the Baikal loss. However it seems that some
potentially critical points have been omitted. The cross-entropy loss is well
known to be beneficial in dataset with severe class imbalance. The two datasets
used for evaluation are perfectly balanced, it might beneficial to see how it
performs in the unbalanced case.

I have a couple of concerns about the method. First about step "(1) loss
function discovery": The initial population starts with trees of depth at most
2, and the final solution(Baikal) has either 2 or 3 (depending on which
definition of depth is chosen). It is unclear that the genetic optimization is
superior to simply choosing random loss functions. I think it would be relevant
to add a figure that shows how the fitness of the leader of each generation
evolves over time.

The second step "(2) coefficient optimization", while objectively generating a
loss function that was superior on the metrics evaluated, raised some
questions. In equation (2) the factor "1.5352" seems to be equivalent to adding
a constant to the loss, which should not impact optimization. Also the factor
"2.7279" seems to be equivalent to a change in learning rate. This may be an
indication that the learning rate search was not done thoroughly. It would be
beneficial to clarify when it is happening: a) For each individual of the
population during step (1), b) before performing CMA, c) after CMA. Also: Was
learning rate search was performed on the network trained with Cross-Entropy? It
was not entirely clear from the experiment details in Appendix A.2.1.

About the Baikal loss itself, I fear that it could produce models that have very
poor calibration, it might be nice to evaluate that (even if it is only in
the appendix).


While the paper does a great job at presenting the problem and its applications
and propose a framework that generated a loss that can transfer to other
datasets without any tuning required. I think it lacks a more thorough
evaluation and description of the dynamics observed during the genetic
evolution, and the performance of the Baikal loss on other datasets (my quick
experients with it on ImageNet diverged I did not have the time necessary to
tune the hyper-parameters).

Minor remarks:

There might be a slight omission in section 3.1: according to Figure 1, exp(x)
is one of the potential unary operators explored by the GLO framework. However
it is not present it the list of operators. Could you clarify this?

To the best of my knowledge, in the machine learning literature, it seems that
the letter x is used to denote the prediction and y for the ground truth. The
fact that this paper used the opposite convention confused me the first time I
read it.


**Experience Assessment:**

I have published one or two papers in this area.

**Review Assessment: Checking Correctness Of Derivations And Theory:**

N/A

**Review Assessment: Checking Correctness Of Experiments:**

I carefully checked the experiments.

**Review Assessment: Thoroughness In Paper Reading:**

I read the paper thoroughly.

---

### Official Review · AnonReviewer1 · 2019-10-22
**Official Blind Review #1**

**Rating:** 3

**Review:**

This paper proposes a very interesting idea of loss function optimization. At first sight, loss function is the goal of optimization and can not be optimized directly. However, the true goal of optimization is the final accuracy (for classification). So lots of loss functions can be designed and combined to form a large search space. In this paper, the authors adopt genetic programming to design loss functions hierarchically.  And experiments show that GLO (Genetic Loss-function Optimization) based loss function can achieve better results than cross entropy.

The paper is well written and easy to understand. I like the idea. Baikal loss is a form searched by GLO. Interestingly and counter intuitively , it is not a monotonically decreasing function. The authors explain it as a regularizer which can prevent the model to be too confident.

Experiments on MNIST and Cifar10 are conducted to show the effectiveness of the proposed method. This part is very weak since MNIST and Cifar10 are very small datasets and the provided results are far from state-of-the-art results. Experiments on larger datasets such as ImageNet and more analysis about the optimization details are suggested to make this work more promising. Since the optimization is rather complex, it's better to show if it is stable enough to generalize to various datasets and models.

**Experience Assessment:**

I have read many papers in this area.

**Review Assessment: Checking Correctness Of Derivations And Theory:**

I assessed the sensibility of the derivations and theory.

**Review Assessment: Checking Correctness Of Experiments:**

I assessed the sensibility of the experiments.

**Review Assessment: Thoroughness In Paper Reading:**

I read the paper at least twice and used my best judgement in assessing the paper.

---

### Official Review · AnonReviewer2 · 2019-10-24
**Official Blind Review #2**

**Rating:** 1

**Review:**

*Summary*
The authors propose using evolutionary computation (EC) to perform meta learning over the set of symbolic expressions for loss functions. It's a compelling idea that is well-motivated. They find that applying their EC method to mnist yields an interesting loss function that they name the 'Baikal loss.' Much of the paper is devoted to analyzing the properties and performance of the Baikal loss.

*Overall Assessment*
The paper's idea is very interesting. However, there are some important drawbacks of this work. These should be fixed and the paper should be resubmitted to a different conference soon.
1) The experiments focus almost entirely on the Baikal loss (a particular loss function found once when running EC on mnist), and do not analyze the overall behavior of EC for loss functions. Does EC consistently converge to the same loss, or do different ones emerge different times you run it? What happens if you optimize convergence speed vs. generalization accuracy with EC? How do these loss functions differ?
2) The experiments are largely on mnist, with a small study showing that the Baikal loss can be applied to cifar-10. It would be good to show that loss functions meta-learned on mnist generalize to larger-scale problems than cifar.

*Comments*
I was surprised when you optimized in fig 3 for convergence speed, rather than final accuracy of something that runs for a while. Why should our goal be to find loss functions that lead to fast optimization, instead of loss functions that lead to models that generalize best? If these are two different goals, then you should have two sets of experiments analyzing how GLO can find interesting (and perhaps different) loss functions for each.

Mnist is possible to get basically 100% accuracy. This means that the loss will only be evaluated in certain regimes of its inputs. What happens when you transfer this to problems where the best achievable accuracy is something like 60% for binary classification?

You should cite the Focal loss as another alternative to the cross entropy loss. Is the focal loss achievable in your particular grammar over loss functions? You should also cite label smoothing as an additional way to achieve a very similar implicit regularization effect as the Baikal loss.

You only analyze one loss function that came from your EC. What if you run it multiple times? Do you find different formulas? How do these perform? The beginning of the paper is very focused on EC, but then you transition suddenly to only discussing the Baikal loss. Can you present experiments demonstrating, for example, how the EC performance varies with the number of steps, with different ways to define the search space, etc?


**Experience Assessment:**

I have published in this field for several years.

**Review Assessment: Checking Correctness Of Derivations And Theory:**

N/A

**Review Assessment: Checking Correctness Of Experiments:**

I assessed the sensibility of the experiments.

**Review Assessment: Thoroughness In Paper Reading:**

I read the paper at least twice and used my best judgement in assessing the paper.

---

### Decision · Program_Chairs · 2019-12-19

**Decision:**

Reject

**Comment:**

This paper proposes a GA-based method for optimizing the loss function a model is trained on to produce better models (in terms of final performance). The general consensus from the reviewers is that the paper, while interesting, dedicates too much of its content to analyzing one such discovered loss (the Baikal loss), and that the experimental setting (MNIST and Cifar10) is too basic to be conclusive. It seems this paper can be so significantly improved with some further and larger scale experiments that it would be wrong to prematurely recommend acceptance. My recommendation is that the authors consider the reviewer feedback, run the suggested further experiments, and are hopefully in the position to submit a significantly stronger version of this paper to a future conference.